# RaCT: Towards Amortized Ranking-Critical Training for Collaborative Filtering

**Sam Lobel**[1†], **Chunyuan Li**[2†*], **Jianfeng Gao**[2], **Lawrence Carin**[3]
[1]Brown University    [2]Microsoft Research, Redmond    [3]Duke University
samuel_lobel@brown.edu   chunyl@microsoft.com

## Abstract

We investigate new methods for training collaborative filtering models based on actor-critic reinforcement learning, to more directly maximize ranking-based objective functions. Specifically, we train a *critic* network to approximate ranking-based metrics, and then update the *actor* network to directly optimize against the learned metrics. In contrast to traditional learning-to-rank methods that require re-running the optimization procedure for new lists, our critic-based method amortizes the scoring process with a neural network, and can directly provide the (approximate) ranking scores for new lists. We demonstrate the actor-critic's ability to significantly improve the performance of a variety of prediction models, and achieve better or comparable performance to a variety of strong baselines on three large-scale datasets.

## 1 Introduction

Recommender systems are an important means of improving a user's web experience. Collaborative filtering is a widely-applied technique in recommender systems (Ricci et al., 2015), in which patterns across similar users and items are leveraged to predict user preferences (Su & Khoshgoftaar, 2009). This naturally fits within the learning paradigm of latent variable models (LVMs) (Bishop, 2006), where latent representations capture the shared patterns. Due to their simplicity and effectiveness, LVMs are still a dominant approach. Traditional LVMs employ linear mappings of limited modeling capacity (Paterek, 2007; Mnih & Salakhutdinov, 2008), and a growing body of literature involves applying deep neural networks (DNNs) to collaborative filtering to create more expressive models (He et al., 2017; Wu et al., 2016; Liang et al., 2018). Among them, variational autoencoders (VAEs) (Kingma & Welling, 2013; Rezende et al., 2014) have been proposed as non-linear extensions of LVMs (Liang et al., 2018). Empirically, VAEs significantly outperform many competing LVM-based methods. One essential contribution to the improved performance is the use of the multinomial likelihood, which is argued by Liang et al. (2018) to be a close proxy to ranking loss.

This property is desirable, because in recommender systems we generally care more about the ranking of predictions than an individual item's score. Hence, prediction results are often evaluated using top-$N$ ranking-based metrics, such as Normalized Discounted Cumulative Gain (NDCG) (Järvelin & Kekäläinen, 2002). The VAE is trained to maximize the likelihood of observations; as shown below, this does not necessarily result in higher ranking-based scores. A natural question concerns whether one may directly optimize against ranking-based metrics, which are by nature non-differentiable and piecewise-constant. Previous work on learning-to-rank has been explored this question in the information-retrieval community, where relaxations/approximations of ranking loss are considered (Weimer et al., 2008; Liu et al., 2009; Li, 2014; Weston et al., 2013).

In this paper, we borrow the actor-critic idea from reinforcement learning (RL) (Sutton et al., 1998) to propose an efficient and scalable learning-to-rank algorithm. The critic is trained to approximate the ranking metric, while the actor is trained to optimize against this learned metric. Specifically, with the goal of making the actor-critic approach practical for recommender systems, we introduce a novel feature-based critic architecture. Instead of treating raw predictions as the critic input, and hoping the neural network will discover the metric's structure from massive data, we consider

---

*Corresponding author   †Equal Contribution

engineering sufficient statistics for efficient critic learning. Experimental results on three large-scale datasets demonstrate the actor-critic's ability to significantly improve the performance of a variety of latent-variable models, and achieve better or comparable performance to strong baseline methods.

## 2 BACKGROUND: VAEs FOR COLLABORATIVE FILTERING

Vectors are denoted as bold lower-case letters $\boldsymbol{x}$, matrices as bold uppercase letters $\mathbf{X}$, and scalars as lower-case non-bold letters $x$. We use $\circ$ for function composition, $\odot$ for the element-wise multiplication, and $|\cdot|$ for cardinality of a set. $\delta(\cdot)$ is the indicator function.

We use $n \in \{1, \ldots, N\}$ to index users, and $m \in \{1, \ldots, M\}$ to index items. The user-item interaction matrix $\mathbf{X} \in \{0,1\}^{N \times M}$ collected from the users' implicit feedback is defined as:

$$x_{nm} \begin{cases} 1, & \text{if interaction of user } n \text{ with item } m \text{ is observed;} \\ 0, & \text{otherwise.} \end{cases} \tag{1}$$

Note that $x_{nm} = 0$ does not necessarily mean user $n$ dislikes item $m$; they may simply be unaware of the item. Further, $x_{nm} = 1$ is not equivalent to saying user $n$ likes item $m$, but that there is at least interest.

**VAE model** VAEs have been investigated for collaborative filtering (Liang et al., 2018), where this principled Bayesian approach is shown to achieve strong performance on large-scale datasets. Given the user's interaction history $\boldsymbol{x} = [x_1, ..., x_M]^\top \in \{0,1\}^M$, our goal is to predict the full interaction behavior with all remaining items. To simulate this process during training, a random binary mask $\boldsymbol{b} \in \{0,1\}^M$ is introduced, with the entry 1 as *un-masked*, and 0 as *masked*. Thus, $\boldsymbol{x}_h = \boldsymbol{x} \odot \boldsymbol{b}$ is the user's partial interaction history. The goal becomes recovering the masked interactions: $\boldsymbol{x}_p = \boldsymbol{x} \odot (1 - \boldsymbol{x}_h)$, which is equivalent to recovering the full $\boldsymbol{x}$ as $\boldsymbol{x}_h$ is known.

In LVMs, each user's binary interaction behavior is assumed to be controlled by a $k$-dimensional user-dependent latent representation $\boldsymbol{z} \in \mathbb{R}^K$. When applying VAEs to collaborative filtering (Liang et al., 2018), the user's latent feature $\boldsymbol{z}$ is represented as a distribution $q(\boldsymbol{z}|\boldsymbol{x})$, obtained from some partial history $\boldsymbol{x}_h$ of $\boldsymbol{x}$. With the assumption that $q(\boldsymbol{z}|\boldsymbol{x})$ follows a Gaussian form, the *inference* of $\boldsymbol{z}$ for the corresponding $\boldsymbol{x}$ is performed as:

$$q_{\boldsymbol{\phi}}(\boldsymbol{z}|\boldsymbol{x}) = \mathcal{N}(\boldsymbol{\mu}, \text{diag}(\boldsymbol{\sigma}^2)), \ \text{ with } \ \boldsymbol{\mu}, \boldsymbol{\sigma}^2 = f_{\boldsymbol{\phi}}(\boldsymbol{x}_h), \ \ \boldsymbol{x}_h = \boldsymbol{x} \odot \boldsymbol{b}, \ \ \boldsymbol{b} \sim \text{Ber}(\alpha), \tag{2}$$

where $\alpha$ is the hyper-parameter of a Bernoulli distribution, $f_{\boldsymbol{\phi}}$ is a $\boldsymbol{\phi}$-parameterized neural network, which outputs the mean $\boldsymbol{\mu}$ and variance $\boldsymbol{\sigma}^2$ of the Gaussian distribution.

After obtaining a user's latent representation $\boldsymbol{z}$, we use the *generative* process to make predictions. In Liang et al. (2018) a multinomial distribution is used to model the likelihood of items. Specifically, to construct $p_{\boldsymbol{\theta}}(\boldsymbol{x}|\boldsymbol{z})$, $\boldsymbol{z}$ is transformed to produce a probability distribution $\boldsymbol{\pi}$ over $M$ items, from which the interaction vector $\boldsymbol{x}$ is assumed to have been drawn:

$$\boldsymbol{x} \sim \text{Mult}(\boldsymbol{\pi}), \ \text{ with } \ \boldsymbol{\pi} = \text{Softmax}(g_{\boldsymbol{\theta}}(\boldsymbol{z})) \tag{3}$$

where $g_{\boldsymbol{\theta}}$ is a $\boldsymbol{\theta}$-parameterized neural network. The output $\boldsymbol{\pi}$ is normalized via a softmax function to produce a probability vector $\boldsymbol{\pi} \in \Delta^{M-1}$ (an $(M-1)$-simplex) over the entire item set.

**Training Objective** Learning VAE parameters $\{\boldsymbol{\phi}, \boldsymbol{\theta}\}$ yields the following generalized objective:

$$\mathcal{L}_{\beta}(\boldsymbol{x}; \boldsymbol{\theta}, \boldsymbol{\phi}) = \mathcal{L}_E + \beta \mathcal{L}_R, \ \text{ with } \ \mathcal{L}_E = -\mathbb{E}_{q_{\boldsymbol{\phi}}(\boldsymbol{z}|\boldsymbol{x})}\big[\log p_{\boldsymbol{\theta}}(\boldsymbol{x}|\boldsymbol{z})\big] \ \text{and} \ \mathcal{L}_R = \text{KL}(q_{\boldsymbol{\phi}}(\boldsymbol{z}|\boldsymbol{x})||p(\boldsymbol{z})) \tag{4}$$

where $\mathcal{L}_E$ is the *negative log likelihood* (NLL) term, $\mathcal{L}_R$ is the KL regularization term with standard normal prior $p(\boldsymbol{z})$, and $\beta$ is a weighting hyper-parameter. When $\beta = 1$, we can lower-bound the log marginal likelihood of the data using equation 4 as $-\mathcal{L}_{\beta=1}(\boldsymbol{x}; \boldsymbol{\theta}, \boldsymbol{\phi}) \leq \log p(\boldsymbol{x})$. This is commonly known as the *evidence lower bound* (ELBO) in variational inference (Blei et al., 2017). Thus equation 4 is the negative $\beta$-regularized ELBO. To improve the optimization efficiency, the *reparametrization trick* (Kingma & Welling, 2013; Rezende et al., 2014) is used to draw samples $\boldsymbol{z} \sim q_{\boldsymbol{\phi}}(\boldsymbol{z}|\boldsymbol{x})$ to obtain an unbiased estimate of the ELBO, which is further optimized via stochastic optimization. We call this procedure *maximum likelihood estimate (MLE)*-based training, as it effectively maximizes the (regularized) ELBO. The testing stage of VAEs for collaborative filtering is detailed in Section A of the Supplement.

**Advantages of VAEs**  The VAE framework successfully scales to relatively large datasets by making use of amortized inference (Gershman & Goodman, 2014): the prediction for all users share the same procedure, which effectively requires evaluating two functions – the encoder $f_\phi(\cdot)$ and the decoder $g_\theta(\cdot)$. Crucially, as all users share the same encoder/decoder, the number of parameters required for an autoencoder is independent of the number of users. This is in contrast to some traditional latent factor collaborative filtering models (Paterek, 2007; Hu et al., 2008; Mnih & Salakhutdinov, 2008), where a unique latent vector is learned for each user. The reuse of encoder/decoder for all users is well-aligned with collaborative filtering, where user preferences are analyzed by exploiting the similar patterns inferred from past experiences (Liang et al., 2018). VAEs has the two advantages *simultaneously*: expressive representation power as a non-linear model, and the number of parameters being independent of the number of users.

**Pitfalls of VAEs**  Among various likelihood forms, it was argued in Liang et al. (2018) that multinomial likelihoods are a closer proxy to the ranking loss than the traditional Gaussian or logistic likelihoods. Though simple and effective, the MLE procedure may still diverge with the ultimate goal in recommendation of correctly suggesting the top-ranked items. To illustrate the divergence between MLE-based training and ranking-based evaluation, consider the example in Figure 1. For the target $\boldsymbol{x} = \{1, 1, 0, 0\}$, two different predictions $A$

| Target | 1 | 1 | 0 | 0 | | NLL | NDCG |
|---|---|---|---|---|---|---|---|
| Prediction A | 0.8 | 0.1 | 0.05 | 0.05 | A | -log 0.08 | 1 |
| Prediction B | 0.3 | 0.3 | 0.35 | 0.05 | B | -log 0.09 | 0.693 |

Figure 1: Difference between MLE-based training loss and ranking-based evaluation. For A, $-1 \times \log 0.8 - 1 \times \log 0.1 = -\log 0.08$; For B, $-1 \times \log 0.3 - 1 \times \log 0.3 = -\log 0.09$. NLL assigns a better value to the misranked example than to the properly-ranked one. NDCG always assigns maximum value to properly-ranked scorings.

and $B$ are provided. In MLE, the training loss is the multinomial NLL: $-\boldsymbol{x} \log \boldsymbol{\pi}$, where $\boldsymbol{\pi}$ is the predicted probability. From the NLL point of view, $B$ is a better prediction than $A$, because $B$ shows a lower loss than $A$. However, $B$ ranks an incorrect item highest, and therefore would return a worse recommendation than $A$. Fortunately, NDCG is calculated directly from the ranking, and so captures this dependence. This inspired us to directly use ranking-based evaluation metrics to guide training. For details on calculating NDCG, refer to Section E.2 of the Supplement.

## 3  RANKING-CRITICAL TRAINING

We introduce a novel algorithm for recommender system training, which we call Ranking-Critical Training (RaCT). RaCT learns a differentiable approximation to the ranking metric, which the prediction network then leverages as a target for optimization through gradient ascent. This is in contrast to existing methods in collaborative filtering, which define an objective relaxation ahead of time. This methodology of learning approximations to functions which cannot be optimized directly stems from the actor-critic paradigm of RL, which we adapt for collaborative filtering.

Any ranking-based evaluation metric can be considered as a "black box" function $\omega : \{\boldsymbol{\pi}; \boldsymbol{x}, \boldsymbol{b}\} \mapsto y \in [0, 1]$, which takes in the prediction $\boldsymbol{\pi}$ to compare with the ground-truth $\boldsymbol{x}$ (conditioned on the mask $\boldsymbol{b}$), and outputs a scalar $y$ to rate the prediction quality. As in equation 2, $\boldsymbol{b}$ partitions a user's interactions into those that are "observed" and "unobserved" during inference. As we are only interested in recovering the unobserved items in recommendation, we compute the ranking score of predicted items $\boldsymbol{\pi}_p = \boldsymbol{\pi} \odot (1 - \boldsymbol{x}_h)$ based on the ground-truth items $\boldsymbol{x}_p$.

One salient component of a ranking-based Oracle metric $\omega^*$ is to sort $\boldsymbol{\pi}_p$. The sorting operation is non-differentiable, rendering it impossible to directly use $\omega^*$ as the critic. While REINFORCE (Williams, 1992) may appear to be suited to tackle the non-differentiable problem, it suffers from large estimate variance (Silver et al., 2014), especially in the collaborative filtering problem, which has a very large prediction space. This motivates consideration of a differentiable neural network to approximate the mapping executed by the Oracle. In the actor-critic framework, the prediction network is called the *actor*, and the network which approximates the oracle is called the *critic*. The actor begins by making a prediction (action) given the user's interaction history as the state. The critic learns to estimate the value of each action, which we define as the task-specific reward, *i.e.,* the Oracle's output. The value predicted by the critic is then used to train the actor. Under the assumption that the critic produces the exact values, the actor is trained based on an unbiased estimate of the gradient of the prediction value in terms of relevant ranking quality metrics. In Figure 2, we illustrate the actor-critic paradigm in (b), and the traditional auto-encoder shown in (a) can be used as the actor in our paradigm.

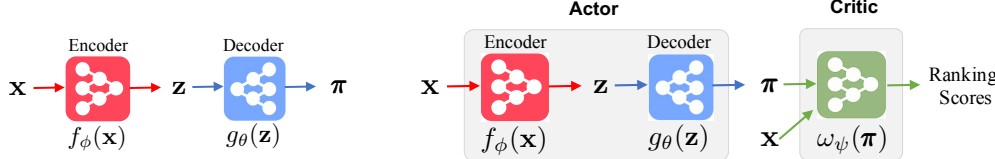

(a) Traditional auto-encoder paradigm      (b) Proposed actor-critic paradigm

Figure 2: Illustration of learning parameters $\{\phi, \theta\}$ in the two different paradigms. (a) Learning with MLE, as in VAEs; (b) Learning with a learned ranking-critic. The *actor* can be viewed as the function composition of encoder $f_\phi(\cdot)$ and $g_\theta(\cdot)$ in VAEs. The *critic* mimics the ranking-based evaluation scores, so that it can provide ranking-sensitive feedback in the actor learning.

**Naive critic**    Conventionally one may concatenate vectors $[\pi_p, x_p]$ as input to a neural network, and train a network to output the measured ranking scores $y$. However, this naive critic is impractical, and failed in our experiments. Our hypothesis is that since this network architecture has a huge number of parameters to train (as the input data layer is of length $2M$, where $M > 10k$), it would require rich data for training. Unfortunately, this is impractical: $\{\pi, x\} \in \mathbb{R}^M$ are very high-dimensional, and the implicit feedback used in collaborative filtering is naturally sparse.

**Feature-based critic**    The naive critic hopes a deep network can discover structure from massive data by itself, leaving much valuable domain knowledge unused. We propose a more efficient critic, that takes into account the structure underlined by the assumed likelihood in MLE (Miyato & Koyama, 2018). We describe our intuition and method below, and provide the justification from the perspective of adversarial learning in Section D of the Supplement.

Consider the computation procedure of the evaluation metric as a function decomposition $\omega = \omega_0 \circ \omega_\psi$, including two steps:

- $\omega_0 : \pi \mapsto h$, feature engineering of prediction $\pi$ into the *sufficient statistics $h$* ;
- $\omega_\psi : h \mapsto \hat{y}$, neural approximation of the mapping from the statistics $h$ to the estimated ranking score $\hat{y}$, using a $\psi$-parameterized neural network.

The success of this two-step critic largely depends on the effectiveness of the feature $h$. We hope feature $h$ is (*i*) *compact* so that fewer parameters in the critic $\omega_\psi$ can simplify training; (*ii*) *easy-to-compute* so that training and testing is efficient; and (*iii*) *informative* so that the necessary information is preserved. We suggest to use a 3-dimensional vector as the feature, and leave more complicated feature engineering as future work. In summary, our feature is

$$h = [\mathcal{L}_E, |\mathcal{H}_0|, |\mathcal{H}_1|], \tag{5}$$

where (*i*) $\mathcal{L}_E$ is the negative log-likelihood in equation 4, defined in the MLE training loss. (*ii*) $|\mathcal{H}_0|$ is the number of unobserved items that a user will interact, with $\mathcal{H}_0 = \{m|x_m = 1 \text{ and } b_m = 0\}$. (*iii*) $|\mathcal{H}_1|$ is the number of observed items that a user has interacted, with $\mathcal{H}_1 = \{m|x_m = 1 \text{ and } b_m = 1\}$.

The NLL characterizes the prediction quality of the actor's output $\pi$ against the ground-truth $x$ in an item-to-item comparison manner, *e.g.,* the inner product between two vectors $-x \log \pi$ as in the multinomial NLL (Liang et al., 2018). Ranking is made easier when there are many acceptable items to rank highly (e.g. when $|\mathcal{H}_0|$ is large), and made difficult when predicting from very few interactions (e.g. when $|\mathcal{H}_1|$ is small), motivating these two features. Including these three features allows the critic to guide training by weighting the NLL's relation to ranking given this context about the user. Interestingly, this idea to consider the importance of user behavior statistics coincides with the scaling trick in SVD (Nikolakopoulos et al., 2019b).

Note that $|\mathcal{H}_0|$ and $|\mathcal{H}_1|$ are user-specific, indicating the user's frequency to interact with the system, which can be viewed as side-information about the user. They are only used as features in training the critic to better approximate the ranking scores, and not in training the actor. Hence, we do not use additional information in the testing stage.

**Actor Pre-training**    In order to be a helpful feature for the critic, the NLL must hold some relationship to the ranking-based objective function. But for the high-dimensional datasets common to collaborative filtering, the ranking score is near-uniformly zero for a randomly-initialized actor. In

this situation, a trained critic will not propagate derivatives to the actor, and therefore the actor will not improve. We mitigate this problem by using a pre-trained actor, such as VAEs that have been trained via MLE.

**Critic Pre-training**  Training a generic critic to approximate the ranking scores for all possible predictions is difficult and cumbersome. Furthermore, it is unnecessary. In practice, a critic only needs to estimate the ranking scores on the restricted domain of the current actor's outputs. Therefore, we train the critic offline on top of the pre-trained MLE-based actor. To train the critic, we minimize the Mean Square Error (MSE) between the critic output and true ranking score $y$ from the Oracle:

$$\mathcal{L}_C(\boldsymbol{h}, y; \boldsymbol{\psi}) = \|\omega_{\boldsymbol{\psi}}(\boldsymbol{h}) - y\|^2, \tag{6}$$

where the target $y$ is generated using its non-differential definition, which plays the role of ground truth simulator in training.

**Actor-critic Training**  Once the critic is well trained, we fix its parameters $\boldsymbol{\psi}$ and update the actor parameters $\{\boldsymbol{\phi}, \boldsymbol{\theta}\}$ to maximize the estimated ranking score

$$\mathcal{L}_A(\boldsymbol{h}; \boldsymbol{\phi}, \boldsymbol{\theta}) = \omega_{\boldsymbol{\psi}}(\boldsymbol{h}), \tag{7}$$

where $\boldsymbol{h}$ is defined in equation 5, including NLL feature extracted from the prediction made in equation 4, together with count features. During back-propagation, the gradient of $\mathcal{L}_A$ wrt the prediction $\boldsymbol{\pi}$ is $\frac{\partial \mathcal{L}_A}{\partial \boldsymbol{\pi}} = \frac{\partial \mathcal{L}_A}{\partial \boldsymbol{h}} \frac{\partial \boldsymbol{h}}{\partial \boldsymbol{\pi}}$. It further updates the actor parameters, with the encoder gradient $\frac{\partial \mathcal{L}_A}{\partial \boldsymbol{\phi}} = \frac{\partial \mathcal{L}_A}{\partial \boldsymbol{\pi}} \frac{\partial \boldsymbol{\pi}}{\partial \boldsymbol{\phi}}$ and the decoder gradient $\frac{\partial \mathcal{L}_A}{\partial \boldsymbol{\theta}} = \frac{\partial \mathcal{L}_A}{\partial \boldsymbol{\pi}} \frac{\partial \boldsymbol{\pi}}{\partial \boldsymbol{\theta}}$. Updating the actor changes its predictions, so we must update the critic to produce the correct ranking scores for its new input domain.

The full RaCT training procedure is summarized in Algorithm 1 in the Supplement. Stochastic optimization is used, where a batch of users $\mathcal{U} = \{\boldsymbol{x}_i | i \in \mathcal{B}\}$ is drawn at each iteration, with $\mathcal{B}$ as a random subset of user index in $\{1, \cdots, N\}$. The pre-training of the actor in Stage 1 and the critic in Stage 2 are important; they provide good initialization to the actor-critic training in Stage 3 for fast convergence. Further, we provide an alternative interpretation to view our actor-critic approach in equation 6 and equation 7 from the perspective of adversarial learning (Goodfellow et al., 2014) in the Supplement. This can partially justify our choice of feature engineering.

## 4  RELATED WORK

**Deep Learning for Collaborative Filtering** There are many recent efforts focused on developing deep learning models for collaborative filtering (Sedhain et al., 2015; Xue et al., 2017; He et al., 2018a;b; Zhang et al., 2017; Chen et al., 2017). Early work on DNNs focused on explicit feedback settings (Georgiev & Nakov, 2013; Salakhutdinov et al., 2007; Zheng et al., 2016), such as rating predictions. Recent research gradually recognized the importance of implicit feedback (Wu et al., 2016; He et al., 2017; Liang et al., 2018), where the user's preference is not explicitly presented (Hu et al., 2008). This setting is more practical but challenging, and is the focus of our work. The proposed actor-critic method belongs to the general two-level architectures for recommendation systems, where a coarse to fine prediction procedure is used. For a systematic method comparison for top-N recommendation tasks, we suggest referring to Dacrema et al. (2019). Our method is closely related to three papers, on VAEs (Liang et al., 2018), collaborative denoising autoencoder (CDAE) (Wu et al., 2016) and neural collaborative filtering (NCF) (He et al., 2017). CDAE and NCF may suffer from scalability issues: the model size grows linearly with both the number of users as well as items. The VAE (Liang et al., 2018) alleviates this problem via amortized inference. Our work builds on top of the VAE, and improves it by optimizing to the ranking-based metric.

**Learned Metrics in Vision & Languages** Recent research in computer vision and natural language processing has generated excellent results, using learned instead of hand-crafted metrics. Among the rich literature of generating realistic images via generative adversarial networks (GANs) (Goodfellow et al., 2014; Radford et al., 2015; Karras et al., 2018), our work is most similar to Larsen et al. (2016), where the VAE objective (Kingma & Welling, 2013) is augmented with the learned representations in the GAN discriminator (Goodfellow et al., 2014) to better measure image similarities. For language generation, the discrepancy between word-level MLE training and sequence-level semantic evaluation has been alleviated with GANs or RL techniques (Bahdanau et al., 2016; Ren et al., 2017; Lin et al.,

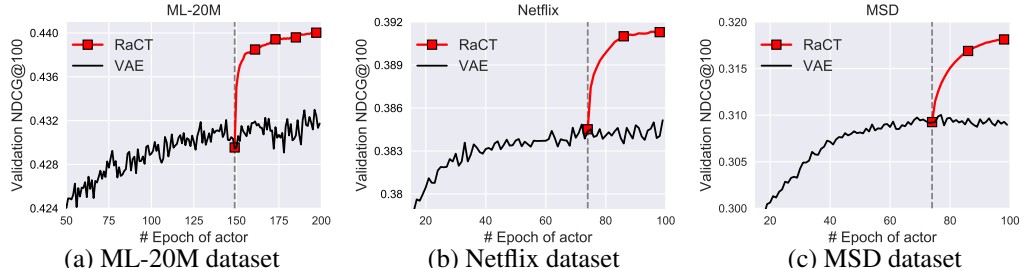

Figure 3: Performance improvement (NDCG@100) with RaCT over the VAE baseline.

2017). The RL approach directly optimizes the metric used at test time, and has shown improvement on various applications, including dialogue (Li et al., 2016), image captioning (Rennie et al., 2017) and translations (Ranzato et al., 2015). Despite the significant successes in other domains, there has been little if any research reported for directly learning the metrics with deep neural networks for collaborative filtering. Our work fills the gap, and we hope it inspires more research in this direction.

**Learning to Rank (L2R)** The idea of L2R has existed for two decades in the information-retrieval community. The goal is to maximize a given ranking-based evaluation metric (Liu et al., 2009; Li, 2014), generally through optimizing objective relaxations (Weimer et al., 2008). Many L2R methods used in recommendation, such as the popular pairwise L2R methods BPR (Rendle et al., 2009) and WARP (Weston et al., 2011), are trained by optimizing a pairwise classification function that penalizes mis-ranked pairs of items. Through negative sampling (Hu et al., 2008), these methods can scale to extremely high-dimensional output spaces. However, it is computationally expensive to compute low-variance updates to a model when the number of items is large.

An alternative to the pairwise approach is *listwise* loss functions, which minimize a loss calculated from a user's entire interaction history. By considering the entire interaction history these methods can more closely model ranking, and generally perform better than their pairwise counterparts (Xia et al., 2008). Furthermore, compared to methods which calculate relative ranking for each pair (Weston et al., 2011), the per-user amortization of rank-calculation can be computed more efficiently. NLL is an example of a listwise loss function, as it is calculated over a user's entire interaction history. Interestingly, NLL is also used as the loss function for ListNet (Cao et al., 2007), a classic listwise L2R method designed to probabilistically maximize Top-1 Recall. The VAE framework under NLL can be seen as a principled extension of this method to Top-N collaborative filtering. Our ranking-critical training further extends this methodology by explicitly calculating the relationship between a differentiable listwise loss function and the desired ranking-based evaluation function.

## 5 EXPERIMENTS

**Experimental Settings** We implemented our algorithm in TensorFlow. The source code to reproduce the experimental results and plots is included as Supplementary Material. We conduct experiments on three publicly available large-scale datasets, which represent different item recommendation scenarios, including user-movie ratings and user-song play counts. This is the same set of user-item consumption datasets used in Liang et al. (2018), and we keep the same pre-processing steps for fair comparison. The statistics of the datasets, evaluation protocols and hyper-parameters are summarized in the Supplement. VAE (Liang et al., 2018) is used as the baseline, which plays the role of our actor pre-training. The NCDG@100 ranking metric is used as the critic's target in training.

**Baseline Methods** We use ranking-critical training to improve the three MLE-based methods described in Section 2.1: VAE, DAE, and MF. We also adapt traditional L2R methods as the actors in our framework, where the L2R loss is used to replace $\mathcal{L}_E$ in equation 5 to construct the feature. We consider WARP and LambdaRank, two pairwise loss functions designed for optimizing NDCG, for these experiments. We also compare our approaches with four representative baseline methods in collaborative filtering. CDAE (Wu et al., 2016) is a strongly-performing neural-network based method, weighted MF (Hu et al., 2008) is a linear latent-factor model, and SLIM (Ning & Karypis, 2011) and EASE (Steck, 2019) are item-to-item similarity models. We additionally compare with Bayesian Pairwise Ranking (Rendle et al., 2009), but as this method did not yield competitive performance on these datasets, we omit the results.

Table 1: Comparison on three large datasets. The best testing set performance is reported. The results below the line are from Liang et al. (2018), and VAE[‡] shows the VAE results based on our runs. Blue indicates improvement over the VAE baseline, and **bold** indicates overall best.

| Dataset | ML-20M | | | Netflix | | | MSD | | |
|---|---|---|---|---|---|---|---|---|---|
| Metric | R@20 | R@50 | NDCG@100 | R@20 | R@50 | NDCG@100 | R@20 | R@50 | NDCG@100 |
| RaCT | **0.403** | **0.543** | **0.434** | 0.357 | **0.450** | 0.392 | 0.268 | 0.364 | 0.319 |
| VAE[‡] | 0.396 | 0.536 | 0.426 | 0.350 | 0.443 | 0.385 | 0.260 | 0.356 | 0.310 |
| WARP | 0.310 | 0.448 | 0.348 | 0.273 | 0.360 | 0.312 | 0.162 | 0.253 | 0.210 |
| LambdaRank | 0.395 | 0.534 | 0.427 | 0.352 | 0.441 | 0.386 | 0.259 | 0.355 | 0.308 |
| EASE | 0.391 | 0.521 | 0.420 | **0.362** | 0.445 | **0.393** | **0.333** | **0.428** | **0.389** |
| VAE | 0.395 | 0.537 | 0.426 | 0.351 | 0.444 | 0.386 | 0.266 | 0.364 | 0.316 |
| CDAE | 0.391 | 0.523 | 0.418 | 0.343 | 0.428 | 0.376 | 0.188 | 0.283 | 0.237 |
| WMF | 0.360 | 0.498 | 0.386 | 0.316 | 0.404 | 0.351 | 0.211 | 0.312 | 0.257 |
| SLIM | 0.370 | 0.495 | 0.401 | 0.347 | 0.428 | 0.379 | – | – | – |

## 5.1 OVERALL PERFORMANCE OF RaCT

**Improvement over VAE** In Figure 3, we show the learning curves of RaCT and VAE on the validation set. The VAE converges to a plateau by the time that the RaCT finishes its actor pre-training stage, *e.g.,* 150 epochs on ML-20 dataset, after which the VAE's performance is not improving. By contrast, when the RaCT is plugged in, the performance shows a significant immediate boost. For the amount of improvement gain, RaCT takes only half the number of epochs that VAE takes in the end of actor pre-training. For example, RaCT takes 50 epochs (from 150 to 200) to achieve an improvement of 0.44-0.43 = 0.01, while VAE takes 100 epochs (from 50 to 150) to achieve an improvement of 0.43-0.424 = 0.006.

**Training/Evaluation Correlation** We visualize scatter plots between learning objectives and evaluation metric for all users on ML-20M dataset in Figure 4. More details and an enlarged visualization is shown in Figure 6 of the Supplement. The Pearson's correlation $r$ is computed. NLL exhibits low correlation with the target NDCG ($r$ is close to zero), while the learned metric in RaCT shows much higher positive correlation. It strongly indicates RaCT optimizes a more direct objective than an MLE approach. Further, NLL should in theory have a negative correlation with the target NDCG, as we

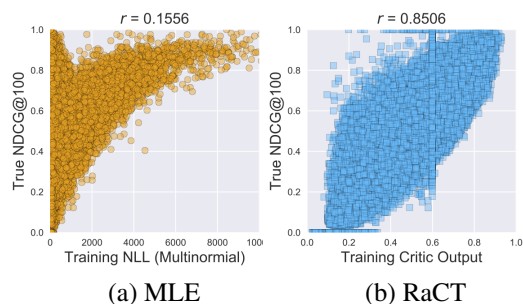

(a) MLE      (b) RaCT

Figure 4: Correlation between the learning objectives (MLE or RaCT) and evaluation metrics on training.

wish that minimizing NLL can maximize NDCG. However, in practice it yields positive correlation. We hypothesize that this is because the number of interactions for each user may dominate the NLL values. That partially motivates us to consider the number of user interactions as features.

**Comparison with traditional L2R methods** As examples of traditional L2R methods, we compare to our method using WARP (Weston et al., 2011) and LambdaRank (Burges et al., 2007) as the ranking-critical objectives. We use implementations of both methods designed specifically to maximize NDCG. We observe that WARP and LambdaRank are roughly 2 and 10 times more computationally expensive than RaCT per epoch, respectively. Table 1 shows the results of RaCT, WARP and LambdaRank, using the same amount of wall-clock training time. We observe the trends that WARP degrades performance, and LambdaRank provides performance roughly equal to VAE. WARP's poor performance is perhaps due to poor approximation of the ranking when the number of items is large.

**Comparison with existing methods** In Table 1, we report our RaCT performance, and compare with competing methods in terms of three evaluation metrics: NDCG@100, Recall@20, and Recall@50. We use the published code[1] of Liang et al. (2018), and reproduce the VAE as our actor pre-training. We further use their reported values for the classic collaborative filtering methods CDAE, WMF, and SLIM. Our reproduced VAE results are very close to Liang et al. (2018) on the ML-20M and

---

[1]`https://github.com/dawenl/vae_cf`

| Actor | Before | After | Gain |
|---|---|---|---|
| VAE | 0.4258 | 0.4339 | 8.09 |
| VAE (Gaussian) | 0.4202 | 0.4224 | 2.21 |
| VAE ($\beta = 0$) | 0.4203 | 0.4255 | 5.17 |
| VAE (Linear) | 0.4156 | 0.4162 | 0.53 |
| DAE (Liang et al., 2018) | 0.4205 | 0.4214 | 0.87 |
| MF (Liang et al., 2018) | 0.4159 | 0.4172 | 1.37 |
| WARP | 0.3123 | 0.3439 | 31.63 |

Table 2: Performance gain ($\times 10^{-3}$) for various actors.

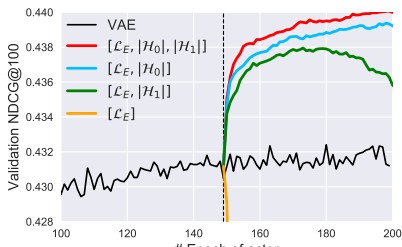

Figure 5: Ablation study on features.

Netflix datasets, but slightly lower on the MSD dataset. The RaCT is built on top of our VAE runs, and consistently improves its baseline actor for all the evaluation metrics and datasets, as seen by comparing the rows RaCT and VAE[‡]. The proposed RaCT also significantly outperforms competing LVMs, including VAE, CDAE, and WMF.

When comparing to EASE (Steck, 2019), our method performs substantially better for ML-20M, comparably for Netflix, and is substantially outperformed for MSD. We observe a similar trend when comparing SLIM (an item-to-item similarity method) and CDAE (a latent variable method). As SLIM and EASE rely on recreating the Gram-matrix $\mathbf{G} = \mathbf{X}^T \mathbf{X}$, their performance should improve with the the number of users (Steck, 2019). However this performance may come at a computational cost, as inference requires multiplication with an unfactored $M \times M$ matrix. EASE requires computing a dense item-to-item similarity matrix, making its inference on MSD roughly 30 times more expensive than for VAE or RaCT. A practitioner's choice between these two methods should be informed by the specifics of the dataset as well as demands of the system.

In the Supplement, we study the generalization of RaCT trained with different ranking-metrics in Section F.1, and break down the performance improvement with different cut-off values of NDCG in Section F.3, and with different number of interactions of $\mathbf{X}$ in Section F.4.

## 5.2 WHAT ACTOR CAN BE IMPROVED BY RACT?

In RL, the choice of policy plays a crucial role in the agent's performance. Similarly, we would like to study how different actor designs impact RaCT performance. Table 2 shows the performance of various policies before and after applying RaCT. The results on NDCG@100 are reported. The VAE, DAE and MF models follow the setup in Liang et al. (2018).

We modify one component of the VAE at a time, and check the change of performance improvement that RaCT can provide. (1) VAE (Gaussian): we change likelihood form from multinomial to Gaussian, and observe a smaller performance improvement. This shows the importance of having a closer proxy of ranking-based loss. (2) VAE ($\beta = 0$): we remove the KL regularization by setting $\beta = 0$, and replace the posterior sampling with a delta distribution. We see a marginally smaller performance improvement. This compares a stochastic and deterministic policy. The stochastic policy (*i.e.,* posterior sampling) provides higher exploration ability for the actor, allowing more diverse samples generated for the critic's training. This is essential for better critic learning. (3) VAE (Linear): we limit the expressive ability of the actor by using a linear encoder and decoder. This significantly degrades performance, and the RaCT cannot help much in this case. RaCT shows improvements for all MLE-based methods, including DAE and MF from Liang et al. (2018). It also shows significant improvement over WARP. Please see detailed discussion in Section F.5 of the Supplement.

## 5.3 ABLATION STUDY ON FEATURE-BASED CRITIC

In Figure 5, we investigate the importance of the features we designed in equation 5, using results from the ML-20M dataset. The full feature vector consists of three elements: $\boldsymbol{h} = [\mathcal{L}_E, |\mathcal{H}_0|, |\mathcal{H}_1|]$. $\mathcal{L}_E$ is mandatory, because it links the actor to the critic; removing it would break the back-propagation to train the actor. We carefully remove $|\mathcal{H}_0|$ or $|\mathcal{H}_1|$ from $\boldsymbol{h}$ at each time, and observe that it leads to performance degradation. In particular, removing $|\mathcal{H}_0|$ results in a severe over-fitting issue. When both counts are removed, we observe an immediate performance drop, as depicted by the orange curve. Overall, the results indicate that all three features are necessary to our performance improvement.

## 6    CONCLUSION & DISCUSSION

We have proposed an actor-critic framework for collaborative filtering on implicit data. The critic learns to approximate the ranking scores, which in turn improves the traditional MLE-based nonlinear LVMs with the learned ranking-critical objectives. To make it practical and efficient, we introduce a few techniques: a feature-based critic to reduce the number of learnable parameters, posterior sampling as exploration for better critic estimates, and pre-training of actor and critic for fast convergence. The experimental results on three large-scale datasets demonstrate the actor-critic's ability to significantly improve the results of a variety of latent-variable models, and achieve better or comparable performance to strong baseline methods.

Though RaCT improves VAEs, it does not start from the best performing actor model. The very recent work by Dacrema et al. (2019) conducts a systematic analysis of algorithmic proposals for top-N recommendation tasks. There are other simple and efficient methods that perform better than VAEs, such as pure SVD-based models (Cremonesi et al., 2010; Nikolakopoulos et al., 2019b), RecWalk (Nikolakopoulos & Karypis, 2019) and Personalized Diffusions (Nikolakopoulos et al., 2019a). One interesting future research direction is to explore learning-to-rank techniques for them.

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

**Summary of contributions:**     Sam and Chunyuan conceptualized learning-to-rank for VAEs. Sam created and implemented the current algorithm, made the model work, and ran all experiments. Chunyuan set up the experiments, led and completed the manuscript writing. Lawrence edited every version of the manuscript. Jianfeng proofread an early version of the manuscript.

## A    TESTING STAGE OF VAES FOR COLLABORATIVE FILTERING

We focus on studying the performance of various models under strong generalization (Liang et al., 2015) as in (Liang et al., 2018). All users are split into training/validation/test sets. The models are learned using the entire interaction history of the users in the training set. To evaluate, we use a part of the interaction history from held-out (validation and test) users to infer the user-level representations from the model, and compute quality metrics by quantifying how well the model ranks the rest of the unseen interaction history from the held-out users. Specifically, for a held-out user with the full history $x$, we take $x_h = x \odot b$ offline using the randomly generated mask $b$. $x_h$ is then frozen as the testing input, and is fed into various trained models during the evaluation stage to get the prediction $\hat{\pi}$. The recovered interaction $\bar{x} = \hat{\pi} \odot (1 - x_h)$ for the masked seen part is then evaluated by ranking-based metrics.

## B    BACKGROUND ON TRADITIONAL LEARNING-TO-RANK METHODS

Formally, the *Bayesian Personalized Ranking* (BPR) (Rendle et al., 2009) loss for the $n$-th user is

$$\mathcal{L}_{\text{BPR}} = \sum_{i \in \mathcal{K}_+} \sum_{j \in \mathcal{K}_-} \sigma(\pi_{nj} - \pi_{ni}), \tag{8}$$

where $\sigma(\cdot)$ is the sigmoid function, $\mathcal{K}_+$ denotes the set of items that the user has interacted with before, and $\mathcal{K}_-$ denotes the complement item set.

The *Weighted Approximate-Rank Pairwise* (WARP) model (Weston et al., 2011) has been shown to perform better than BPR for implicit feedback (Kula, 2015):

$$\mathcal{L}_{\text{WARP}} = \sum_{ni \in \mathcal{K}_+} \sum_{j \in \mathcal{K}_-} w(r_i)\text{max}(0, 1 + \pi_{nj} - \pi_{ni})), \tag{9}$$

where $w(\cdot)$ is a weighting function for different ranks, and $r_i$ is the rank for the $i$-th item for $n$-th user. A common choice of weighting function $w(\cdot)$ for optimizing NDCG is $w(r) = \sum_{i=1}^{r} \alpha_i$, with $\alpha_i = 1/i$. WARP improves BPR by the weights $w(\cdot)$ and the margin between positive and negative items.

## C    PSEUDO-CODE FOR RACT

We summarize the full training procedure of RaCT in Algorithm 1.

## D    INTERPRETATION WITH GANS

We can view our actor-critic approach in equation 6 and equation 7 from the perspective of Generative Adversarial Networks (GANs). GANs constitute a framework to construct a *generator G* that can mimic a target distribution, and have achieved significant success in generating realistic images (Goodfellow et al., 2014; Radford et al., 2015; Karras et al., 2018; Brock et al., 2018). The most distinctive feature of GANs is the *discriminator D* that evaluates the divergence between the current generator distribution and the target distribution (Goodfellow et al., 2014; Li et al., 2017). The GAN learning procedure performs iterative training between the discriminator and generator, with the discriminator acting as an increasingly meticulous critic to refine the generator. In our work, the actor can be interpreted as the generator, while the critic can be viewed as the discriminator.

Note that GANs and actor-critic models learn the metric functions (Finn et al., 2016), and it has been shown in Pfau & Vinyals (2016) that GANs can be viewed as actor-critic in an environment where the actor cannot affect the reward. This is exactly our setup. One key difference is that we know the Oracle metric, and the critic is trained to mimic the Oracle's behaviour.

---

**Algorithm 1:** Our full ranking-critical training with stochastic optimization.

---

**Input :** Interaction matrix $\mathbf{X}$; Actor parameters (encoder $\phi$ and decoder $\theta$), Critic parameters $\psi$.

1 **Initialize**: Randomly initialize weights $\phi$, $\theta$ and $\psi$

2 /* Stage 1:  Pretrain  the  actor  via  MLE                          */

3 **while** *not converged do* **do**

4     Sample a batch of users $\mathcal{U}$;

5     Update $\{\theta, \phi\}$ with gradient $\frac{\partial \mathcal{L}_\beta}{\partial \theta}$ and $\frac{\partial \mathcal{L}_\beta}{\partial \phi}$ in equation 4;

6 **end**

7 /* Stage 2:  Pretrain the critic via MSE                             */

8 **while** *not converged do* **do**

9     Sample a batch of users $\mathcal{U}$;

10     Construct features $\boldsymbol{h}$ in equation 5 and target $y$ from the Oracle;

11     Update $\psi$ with gradient $\frac{\partial \mathcal{L}_C}{\partial \psi}$ in equation 6;

12 **end**

13 /* Stage 3:  Alternative training of actor and critic                */

14 **for** $t = 1, 2, \ldots, T$ **do**

15     Sample a batch of users $\mathcal{U}$;

16     /*  Actor   step                                              */

17     Update $\{\theta, \phi\}$ with gradient $\frac{\partial \mathcal{L}_A}{\partial \theta}$ and $\frac{\partial \mathcal{L}_A}{\partial \phi}$ in equation 7;

18     /*  Critic   step                                             */

19     Construct features $\boldsymbol{h}$ in equation 5 and target $y$ from the Oracle;

20     Update $\psi$ with gradient $\frac{\partial \mathcal{L}_C}{\partial \psi}$ in equation 6;

21 **end**

---

Conditioned on interaction history $\boldsymbol{x}_h$ corrupted from $\boldsymbol{x}$, the actor predicts the distribution parameter $\boldsymbol{\pi}$ over items, which further constructs the likelihood $p(\boldsymbol{x}|\boldsymbol{\pi})$. We use $q$ to designate the data empirical distribution, the target conditional is $q(\boldsymbol{x}|\boldsymbol{\pi})$. It can be formulated as the standard adversarial loss for the conditional GAN (Mirza & Osindero, 2014). It has been shown that the optimal critic (Goodfellow et al., 2014; Li et al., 2017) for a conditional GAN can be represented as the log likelihood ratio

$$R^*(\boldsymbol{\pi}, \boldsymbol{x}) = \log \frac{q(\boldsymbol{x}|\boldsymbol{\pi})}{p(\boldsymbol{x}|\boldsymbol{\pi})} \tag{10}$$

In the collaborative filtering setup, we often make the assumptions that $p(\boldsymbol{x}|\boldsymbol{\pi})$ are simple distributions, such as multinomial in VAEs (Liang et al., 2018) and Gaussian in MF. This simplification allows the parameterization of critic following the following form (Miyato & Koyama, 2018):

$$R^*(\boldsymbol{\pi}, \boldsymbol{x}) = \boldsymbol{x}^\top \mathbf{V} \nu(\boldsymbol{\pi}) + \mathbf{C} \tag{11}$$

where $\boldsymbol{x}$ is the target, $\nu(\boldsymbol{\pi})$ is a layer of the critic with input $\boldsymbol{\pi}$, and $\mathbf{V}$ and $\mathbf{C}$ are the parameters to learn. Most notably, this formulation introduces the prediction information via an inner product, as opposed to concatenation. The form equation 11 is indeed the form we proposed for NLL feature $\boldsymbol{x} \log \boldsymbol{\pi}$, with $\mathbf{V} = \mathbf{I}$ and $\nu(\cdot) = \log(\cdot)$. $\mathbf{C}$ includes the normalizer for the prediction probability (Miyato & Koyama, 2018), which is related to the count features in equation 5.

## E   EXPERIMENTAL SETUP

### E.1   DATASETS

We conduct experiments on three publicly available datasets. Table 3 summarizes the statistics of the data. These three ten-million-size datasets represent different item recommendation scenarios, including user-movie ratings and user-song play counts. This is the same set of medium- to large-scale user-item consumption datasets used in Liang et al. (2018), and we keep the same pre-processing steps for fair comparison.

Table 3: Summary Statistics of datasets after all pre-processing steps. *Interactions#* is the number of non-zero entries. *Sparsity%* refers to the percentage of zero entries in the user-item interaction matrix $X$. *Items#* is the number of total items. *HO#* is the number of validation/test users held out of the total number of users in the 5th column *Users#*.

| Dataset | Interaction# | Sparsity% | Item# | User# | HO# |
|---------|-------------|-----------|-------|-------|-----|
| ML-20M | 10.0M | 99.64% | 20,108 | 136,677 | 10K |
| Netflix | 56.9M | 99.31% | 17,769 | 463,435 | 40K |
| MSD | 33.6M | 99.86% | 41,140 | 571,355 | 50K |

1. **MovieLens-20M (ML-20M)**: This is the user-movie rating data collected from a movie recommendation service[2]. The data is binarized by keeping ratings of four or higher and setting other entries as unobserved. Only users who have watched at least five movies are considered.

2. **Netflix Prize (Netflix)**: This is the user-movie rating data from the Netflix Prize[3]. Similarly to ML-20M, the data is binarized by keeping ratings of four or higher, and only users who have watched at least five movies are kept.

3. **Million Song Dataset (MSD)**: This is the user-song play count data from the Million Song Dataset (Bertin-Mahieux et al., 2011). We binarize play counts, and keep users who have listened to at least 20 songs as well as songs that are listened to by at least 200 users.

### E.2 EVALUATION PROTOCOL

In the testing stage, we get the predicted ranking by sorting the multinomial probability $\pi_p$. For each user, we compare the predicted ranking of the held-out items with their true ranking. Two ranking-based metrics are considered, Recall@R and the truncated NDCG (NDCG@R), where $R$ is the cut-off hyper-parameter. While Recall@R considers all items ranked within the first $R$ to be equally important, NDCG@R uses a monotonically increasing discount to emphasize the importance of higher ranks versus lower ones.

Formally, we define $m(r)$ as the item at rank $r$, and $\mathcal{H}_0$ as the held-out unobserved items that a user will interact.

$$\text{DCG@R} = \sum_{r=1}^{R} \frac{2^{\delta[m(r) \in \mathcal{H}_0]} - 1}{\log(r+1)}. \tag{12}$$

By dividing DCG@R by its best possible value, we obtain NDCG@R in $[0, 1]$.

$$\text{Recall@R} = \sum_{r=1}^{R} \frac{\delta[m(r) \in \mathcal{H}_0]}{\min(R, |\mathcal{H}_0|)}. \tag{13}$$

The denominator normalizes Recall@R in $[0, 1]$, with maximum value 1 corresponding to the case that all relevant items are ranked in the top $R$ positions.

### E.3 EXPERIMENT HYPER-PARAMETERS

We set hyper-parameters by following Liang et al. (2018) for comparisons. For VAE, the dimension of the latent representation is 200. When KL regularization is removed ($\beta = 0$), *i.e.,* for DAE and MF, we instead apply $\ell_2$ regularization (0.01) on weights to prevent overfitting. Adam optimizer (Kingma & Ba, 2014) is used, with batch size of $|\mathcal{B}| = 500$ users. For ML-20M, the actor is pre-trained for 150 epochs, and alternative training for 50 epochs. On the other two datasets, the actor is pre-trained for 75 epochs, and alternative training for 25 epochs. The critic is pre-trained for 50 epochs for all three datasets. The alternative training has equal update frequency for actor and critic. This schedule ensures that we the have the same total number of actor training epochs as Liang et al. (2018): 200 epochs for ML-20M, 100 epochs for the other two datasets.

---

[2]https://grouplens.org/datasets/movielens/20m/
[3]https://www.netflixprize.com/

Table 4: Network architectures. The arrow indicates the flow between two layers. For each layer, we show the number of units on top of its following activation function. BN indicates Batch Normalization.

| Networks | | Architectures | | | |
|---|---|---|---|---|---|
| Actor | Encoder | $\underset{\text{Linear}}{M} \to \underset{\text{Tanh}}{600} \to \underset{\text{Linear\& Exp}}{200}$ | | | |
| | Decoder | $\underset{\text{Linear}}{200} \to \underset{\text{Tanh}}{600} \to \underset{\text{Softmax}}{M}$ | | | |
| Critic | | $\underset{\text{BN}}{3} \to \underset{\text{ReLU}}{100} \to \underset{\text{ReLU}}{100} \to \underset{\text{ReLU}}{10} \to \underset{\text{Sigmoid}}{1}$ | | | |

Table 5: Summary of training schedule hyper-parameters. $\beta_{\max}$ indicates the maximum value of $\beta$. In the actor pre-training stage, the number of epochs used for increasing and fixing $\beta$ are shown in row 3 and 4, respectively.

| Dataset | ML-20M | Netflix | MSD |
|---|---|---|---|
| $\beta_{\max}$ | 0.2 | 0.2 | 0.1 |
| # epochs for annealing | 100 | 75 | 75 |
| # epochs for fixing | 50 | 0 | 0 |
| # epochs for actor pre-training | 150 | 75 | 75 |
| # epochs for critic pre-training | 50 | 50 | 50 |
| # epochs for alternative training | 50 | 25 | 25 |

Table 6: Performance trained with different metrics. (ML-20M)

| Training | Testing | | |
|---|---|---|---|
| | Recall@20 | Recall@50 | NDCG@100 |
| RaCT (Recall@100) | **0.40316** | **0.54317** | 0.43392 |
| RaCT (NDCG@100) | 0.40269 | 0.54304 | **0.43395** |
| VAE | 0.39623 | 0.53632 | 0.42586 |

A fully-connected (FC) architecture is used for all networks, as detailed in Table 4. Please refer to Goodfellow et al. (2016) for the activation functions. Batch Normalization (Ioffe & Szegedy, 2015) is used to normalize the input features, because the magnitude of the inputs (NLL) change as training progresses. The encoder outputs the mean and variance of the varational distribution; the variance is implemented via an exponential function.

## F    ADDITIONAL EXPERIMENTAL RESULTS

### F.1    GENERALIZATION ACROSS RANKING METRICS

To study the generalization ability of RaCT, we consider training the critic against Recall@100, in addition to NDCG@100. The only difference is that Recall treats each item as equally important, while NDCG treats the higher ranking items as more important. The results are shown in Table 6. Indeed, the RaCT gets slightly better testing Recall values when trained against the Recall metric, and the reverse holds for NDCG. More importantly, RaCT allows generalization across different ranking metrics: all testing metric values are significantly improved when trained against either Recall or NDCG.

Following Liang et al. (2018), we compare with NCF on two small datasets, ML-1M (6,040 users, 3,704 items) and Pinterest (55,187 users, 9,916 items). This is because the prediction stage of NCF is slow, due to a lack of amortized inference as in VAE. We use their publicly available datasets and metrics for fair comparison. The results are evaluated with a small cut-off value $R$, to only study the highly ranked items: NDCG@10 and Recall@10. The performance are compared in Table 7. Our observation that DAE performs better than VAE on these two datasets is consistent with Liang

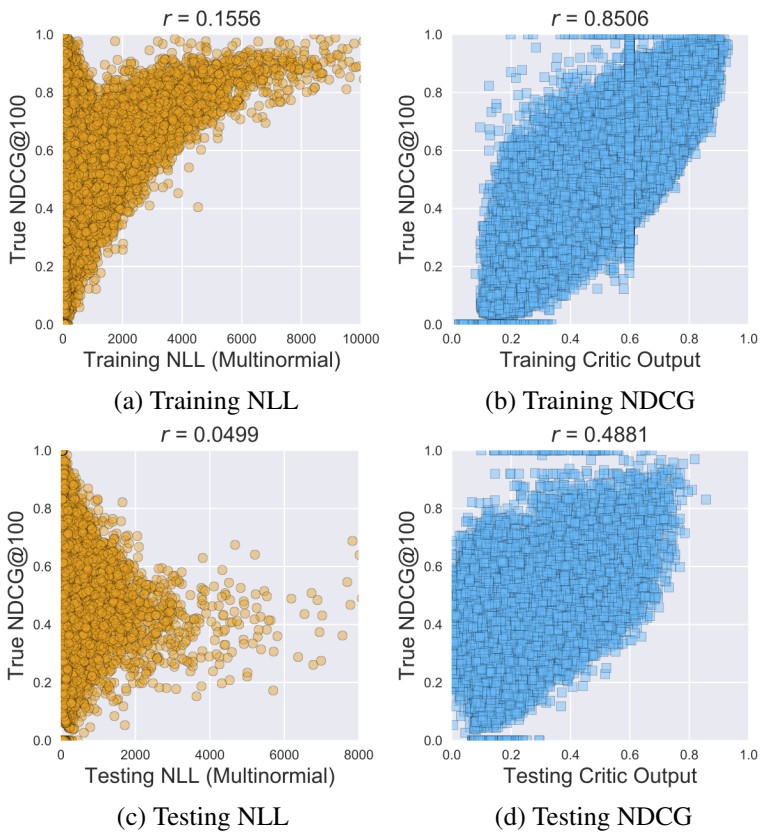

Figure 6: Correlation between the learning objectives (NLL or RaCT) and evaluation metrics NDCG.

Table 7: Comparison between our RaCT with NCF on two small datasets. NCF results are from Liang et al. (2018).

| Dataset | Metric | NCF | DAE | RaCT | VAE | RaCT |
|---------|--------|-----|-----|------|-----|------|
| ML-1M | Recall@10 | 0.705 | 0.722 | 0.722 | 0.704 | 0.706 |
| | NDCG@10 | 0.426 | 0.446 | 0.446 | 0.433 | 0.434 |
| Pinterest | Recall@10 | 0.872 | 0.886 | 0.887 | 0.873 | 0.878 |
| | NDCG@10 | 0.551 | 0.580 | 0.581 | 0.564 | 0.568 |

et al. (2018). In general, RaCT shows higher improvement when a larger dataset (Pinterest), or a stochastic actor (VAE) is considered. This is because the sizes of the two datasets are relatively small, the critic can be better trained when more samples are observed. On the larger Pinterest dataset, the auto-encoder variants perform better than NCF by a big margin, and our RaCT further boosts the performance.

## F.2 CORRELATION BETWEEN TRAINING METRICS

Figure 6 explores the relationship between NLL, the learned RaCT metric, and NDCG. We ensure that the best model for each method is used: the model after actor pre-training (Stage 1) is used for NLL plots, and the model after the actor-critic alternative training (Stage 3) is used for RaCT plots. The bottom row of plots displays the output of these models on the testing data. This demonstrates that RaCT's connection to the ranking metric generalizes to unseen data.

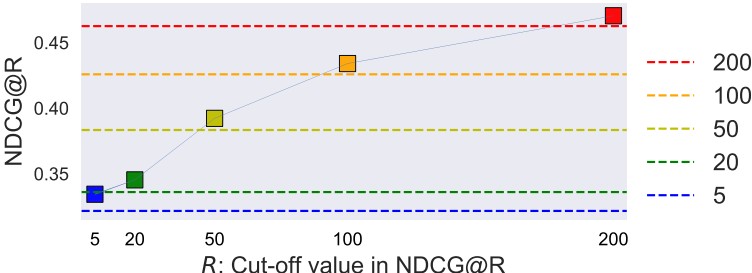

Figure 7: The improvement at various cut-off value R in evaluation. Given a specific R, the dashed line shows the VAE, and square dot shows the RaCT.

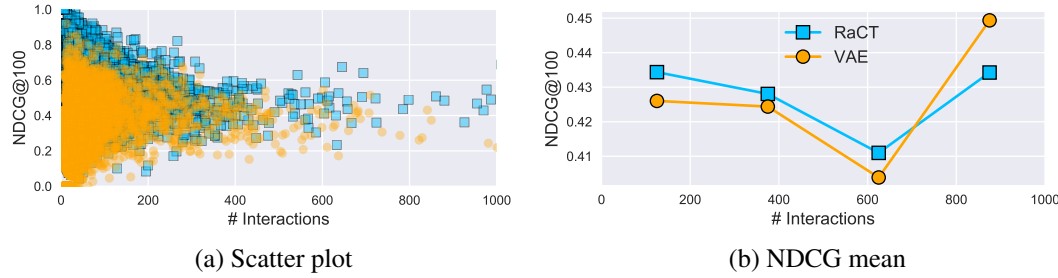

(a) Scatter plot  (b) NDCG mean

Figure 8: Improvement breakdown over different user interactions. (a) Scatter plot between NDCG@100 and activity levels. Note only # interactions $\leq 1000$ is visualized, there is a long tail (>1000) in the distribution. (b) Comparison of the mean NDCG@100 values for four user groups.

### F.3 BREAKDOWN ANALYSIS FOR DIFFERENT CUT-OFF VALUES

NDCG@100 only reflects the ranking quality at the cut-off value $R = 100$. *i.e.,* the top-100 ranking items. To study the ranking quality at different range of the predicted list, we consider a large range of $R$, and report the corresponding NDCG values. We consider $R = 5, 20, 50, 100, 200$, and report the results in Figure 7. The NDCG@R values are improved for various R, though the critic is trained against NDCG@100. This is because the NDCG metrics of different R are highly correlated, the RaCT can generalize across them.

### F.4 BREAKDOWN ANALYSIS FOR DIFFERENT NUMBER OF INTERACTIONS

In Figure 8, we show performance improvement across increasing user interactions. We use ML-20M dataset for this case study. The # interactions is the number of items each user interacts with (ground-truth), indicating the user's activity level. Figure 8(a) shows the scatter plots between NDCG@100 values and various number of interactions on the testing dataset, for both VAE and our RaCT methods. RaCT generally improves VAE for a large range of user interactions. We further categorize the users in four groups according to their number of interactions: $<250, 250-500, 501-750, >750$, and plot the mean of NDCG@100 values for two methods in Figure 8(b). RaCT improves VAE except for users with high activity level (>750). This is probably because the number of the most active users is small, as observed in Figure 8(a). It yields a lack of training data for critic learning, which potentially hurts the performance.

### F.5 ON THE PERFORMANCE IMPROVEMENT OF ACTORS VIA RaCT.

We also consider the two other auto-encoder variants used in Liang et al. (2018) as the actor. (1) The DAE in Liang et al. (2018) chooses a smaller architecture $M \to 600 \to M$, which achieves better performance than the larger architecture as in our VAE ($\beta = 0$) by prevent over-fitting. While we observe the same result, it is interesting to note that the VAE ($\beta = 0$) shows a much larger improvement gain than DAE (Liang et al., 2018) when trained with our RaCT technique, and eventually significantly outperforms the latter. This shows that the additional modeling capacity is

necessary to capture the more complex relationship in prediction, when the goal is ranking rather than MLE. (2) The MF in Liang et al. (2018) employs a Gaussian likelihood, which also gets slight improvement with the RaCT. Overall, we can conclude that the RaCT method improves all the MLE-based variants.

We also use ranking-loss-based WARP as the actor. For the large datasets considered in this paper, calculating the full WARP-loss for each user is impractically slow. We derive a simple approximation to WARP which runs in quasilinear time to the number of items. Even so, it takes around 30 minutes per epoch on ML-20M dataset, roughly 30 times slower than the VAE. WARP yields the score 0.312, which is lower than other baseline methods. This is consistent with the studies in Liang et al. (2018); Sedhain et al. (2016). However, when RaCT is applied, WARP gets a significant improvement; in fact, the largest improvement gain of all the actors. This indicates the RaCT is a more direct and effective approach for learning to rank on large datasets.

