# OpenReview forum: "RaCT: Toward Amortized Ranking-Critical Training For Collaborative Filtering "
_ICLR.cc/2020/Conference — Accept (Poster)_

### Official Review · AnonReviewer2 · 2019-10-16
**Official Blind Review #2**

**Rating:** 8

**Review:**

The authors propose a novel reinforcement learning (RL) based recommendation algorithm (i.e., Ranking-Critical Training, RaCT) for modeling users' implicit feedback, which is illustrated in Figure 2(b). Specifically, the authors apply the actor-critic RL paradigm to approximately optimize the ranking-oriented loss/objective function in collaborative filtering with implicit feedback, which is a difficult and important problem. In particular, the critic network is used to approximate the ranking-oriented metric, and the actor network is used for optimizing this metric. Moreover, for efficiency, the authors also propose a feature-based critic to replace the original one.

Empirical studies on three large datasets, i.e., ML20M, Netflix and MSD, show very promising results in comparison with the state-of-the-art methods on the studies problem. The results are convincing.

The paper is well presented, including clear background and good illustration, etc.

The proposed model is believed to be interested by the community of both recommender systems and reinforcement learning. I thus recommend acceptance.


**Experience Assessment:**

I have published in this field for several years.

**Review Assessment: Checking Correctness Of Derivations And Theory:**

I assessed the sensibility of the derivations and theory.

**Review Assessment: Checking Correctness Of Experiments:**

I carefully checked the experiments.

**Review Assessment: Thoroughness In Paper Reading:**

I read the paper at least twice and used my best judgement in assessing the paper.

---

> ### Author Response · Authors · 2019-11-14
> **Response to Reviewer #2**
>
> We appreciate your supportive review.

---

### Official Review · AnonReviewer3 · 2019-10-23
**Official Blind Review #3**

**Rating:** 6

**Review:**

The work presents a two-level architecture for building an efficient ranking solution for standard collaborative filtering task in recommender systems. The core of this solution is the actor-critic approach, where the critic (motivated by ideas from RL) tries to directly approximate a ranking-based metric using as an input prediction from the actor (VAE framework) and the ground truth knowledge from an Oracle. Both actor and critic require a pre-training step, after which the critic is set to propagate adjustments to actor parameters. Experiments demonstrate the advantage of such approach over competing methods.

The work presents an interesting idea of feature-based critic for learning a better ranking. The choice of features is surprisingly effective considering how simple they are – just basic statistics. On the other hand, the motivating example with NLL vs NDCG is not convincing. Comparing to much simpler linear SOTA methods, the improvement over multinomial VAE is not impressive. This is probably due to VAE being not a real SOTA in the first place, however other possible candidates for actor were not demonstrated. Anyway, I believe that this is an important direction for research and can be considered for acceptance if the authors adjust the text according to the comments below.

Main arguments
The text is well structured, easy to read. Key concepts are explained carefully with the appropriate level of details. Open-source repository with the source code is definitely a plus. However, there’s a number of inaccuracies, which should be addressed.

In the example demonstrating VAE's pitfalls, it is stated that prediction A is intuitively better than prediction B. I’d argue with that intuition. For me, B looks more consistent as there’s a smaller gap between predictions for positive targets. In A one of the predictions is much lower, whcih seems more like an outlier. Moreover, shouldn’t NDCG be 1 in both cases? The ordering of positive examples is the same in both cases. Please, consider addiing more explanations to clarify your intuition or replace the example with something less contradicting.

The  statement “making RaCT scalable to large-scale datasets” is arguable from the technical viewpoint. As far as I know, there’s still no good support for sparse calculations in NN frameworks. When you take user mini batch you have to convert it to a dense matrix with explicitly stored zeros. It’s not a problem for the datasets tested in the paper (with <100K items). However, consider 100M items. What will be the size of user batch that will fit GPU memory? Will it be still large enough and effective from the learning viewpoint (smaller batches are likely to reduce generalization)? As it will mostly contain zeroes (few know interactions vs ~100M unknown), efficiency of calculations with this batch will also be low comparing to operations in sparse format. In that sense the datasets used in the paper are not really large scale.
Another false statement: “This is much more efﬁcient than most traditional latent factor collaborative ﬁltering models …, where a time-consuming optimization procedure is typically performed to obtain the latent factor for a user who is not present in the training data”. All factorization models admit folding-in procedure, which doesn’t depend on the size of the problem (i.e., the number of users or items) and depends only on the rank of decomposition (the number of latent variables). This is among the reasons why MF is so popular in industrial applications.

Furthermore, SVD-based models (I’m talking about the real SVD, not MF models like SVD++) are absolute winners here as they do not even require running any additional optimization steps. Due to the orthogonality property, the solution for folding-in has an analytical form in the case of SVD, e.g., given a low-rank item embeddings matrix V with orthonormal columns, the folding-in solution (vector of predicted item scores for a user) is simply given by an orthogonal projector
r = VV^T p,
for any vector of user preferences p, be it a known user or a new one (i.e., not present in the training data). It is basically a simple encoder-decoder architecture with a single linear layer. Training SVD-based models can also very efficient and can be scaled to almost billion-size problems, e.g. https://github.com/criteo/Spark-RSVD. Moreover, one can significantly improve standard PureSVD model [Cremonesi, Koren, Turrin 2010] by a simple scaling trick as it is shown in the EigenRec model [Nikolakopoulos 2019], you can also see more details on this in the blog https://www.eigentheories.com/blog/to-svd-or-not-to-svd. I also find it interesting that the scaling trick in SVD partially intersects with the idea of importance of user behavior statistics in the critic model,  as it also depends on how many clicks a user has made.

I ran my own experiments with VAE and SVD in google colab (I can provide a link to the notebook if needed). With a proper tuning the SVD-based solution gives NDCG=0.415, which is within 3% difference of the Mult-VAE’s best result on the ML-20M dataset. It takes around 200s to train the SVD model of rank 600 on the CPU in the google colab. For comparison, one epoch of VAE training takes 26s in the same google colab environment (which I believe is using Tesla K80 GPU). The same NDCG level is achieved by Mult-VAE after approximately 20 epochs, which amounts to more than 500s of training time, twice longer than SVD. Speaking about practicality, even a student will have a decent CPU nowadays, but not everyone will have Tesla K80. SVD merely requires doing `from scipy.sparse.linalg import svds` in most cases. Moreover, there are even better solutions than SVD (in terms of accuracy), still linear, not even based on MF, e.g., RecWalk [Nikolakopoulos and Karypis 2019] and even better one - Personalized Diffusions model [Nikolakopoulos, Berberidis, Karypis, Giannakis 2019] - that are shown to significantly outperform VAE models.

The general conclusion is that even though you improve VAE, you are actually starting from not the best model and there’s much more room for improvement (demonstrated by much simpler models) than you show in your work. Please consider this fact and adjust wording in your text accordingly. Especially in the statements like “VAEs signiﬁcantly outperform many state-of-the-art methods” or “the proposed RaCT also signiﬁcantly outperforms other state-of-the-art methods, including VAE, CDAE, WMF and SLIM” as it is really about outperforming strong baselines not real SOTA.
In addition to that, recent studies show that many modern neural networks-based approaches underperform even simple KNN-based models, see work by [Dacrema, Cremonesi, Jannach 2019] on “A Worrying Analysis of Recent Neural Recommendation Approaches”. It includes VAE, DAE and NeuCF that you reference in your paper.

In spite of that, the statement “[LVM]… may yield suboptimal performance, especially for large datasets (He et al., 2017)” is at best questionable and requires more investigation as the referenced here paper is about NeuCF. This also makes me doubt in the claims, where you talk about approximation of the real ranking measure by the critic. Examples with RecWalk and PersDiff show that even without optimizing any ranking metric you can get a decent quality in terms of NDCG. Even though I find the general explanation in the text intuitive and appealing, I believe, more rigorous research should be done before we can have something reliable to say about ranking metric approximation. I’d suggest being more careful with such statement in the text. Even in your own experiments a simple EASE model is sometimes better, even though it has nothing to do with approximating any elaborate ranking metric (it's closer to the nearest neighbours search).


Other comments
1)	“performance comes at a computational cost, as inference requires multiplication with an unfactored M ×M matrix” is arguable to a certain extent. In the case of SLIM there’s a sparse matrix with zero diagonal, so once it’s computed and stored in a sparse format, it’s not really a problem to do calculations with it (the training is expensive, though).
2)	 “which is argued to be a close proxy to ranking loss” a reference required here.
3)	The two-level architectures are very popular in general. At the first level, a compact list of candidates is generated by a quick to compute algorithm like MF,  then the second-level algorithm based on decision-trees or multiarmed bandits (MAB) is trained on a much smaller output space. Comparing with such approaches (especially with MAB as they are widely used in industry, probably the second popular approach after MF) would make the work more complete.
4)	“Previous work on learning-to-rank has been explored this question…” something is not right here.
5)	Unfortunately, a lot of prior studies do not report confidence intervals, and this paper is not an exception. It makes it hard to reason about statistical significance of the results and about generalization capabilities of the approach.

References:
Cremonesi, Paolo, Yehuda Koren, and Roberto Turrin. "Performance of recommender algorithms on top-n recommendation tasks." In Proceedings of the fourth ACM conference on Recommender systems, pp. 39-46. ACM, 2010.
Nikolakopoulos, Athanasios N., Vassilis Kalantzis, Efstratios Gallopoulos, and John D. Garofalakis. "EigenRec: generalizing PureSVD for effective and efficient top-N recommendations." Knowledge and Information Systems 58, no. 1 (2019): 59-81.
Nikolakopoulos, Athanasios N., and George Karypis. "Recwalk: Nearly uncoupled random walks for top-n recommendation." In Proceedings of the Twelfth ACM International Conference on Web Search and Data Mining, pp. 150-158. ACM, 2019.
Nikolakopoulos, Athanasios N., Dimitris Berberidis, George Karypis, and Georgios B. Giannakis. "Personalized diffusions for top-n recommendation." In Proceedings of the 13th ACM Conference on Recommender Systems, pp. 260-268. ACM, 2019.
Dacrema, Maurizio Ferrari, Paolo Cremonesi, and Dietmar Jannach. "Are we really making much progress? A worrying analysis of recent neural recommendation approaches." In Proceedings of the 13th ACM Conference on Recommender Systems, pp. 101-109. ACM, 2019.

**Experience Assessment:**

I have published in this field for several years.

**Review Assessment: Checking Correctness Of Derivations And Theory:**

I assessed the sensibility of the derivations and theory.

**Review Assessment: Checking Correctness Of Experiments:**

I carefully checked the experiments.

**Review Assessment: Thoroughness In Paper Reading:**

I read the paper thoroughly.

---

> ### Author Response · Authors · 2019-11-14
> **Response to Reviewer #3**
>
> Thank you for your detailed and thoughtful review.
>
> See above for corrections to our NLL-vs-NDCG example — we have updated the PDF to better contrast the two.
>
> Q: On the statement “Making RaCT scalable to large-scale datasets”:
> A: We’d like to clarify that our scalability means that the estimate of ranking and gradient computation happens once per user, in contrast to pairwise methods such as WARP which require an estimate for each pair of items. We have adjusted our wording to emphasize ``efficiency’’ rather than ``scalable to large-scale datasets’’,  . This is moved this argument to the “Learning To Rank” section for clarity (“Furthermore, compared to methods...”).
>
> Q: On “folding-in” for fast inference on new users:
> A: Thank you, we were unaware of this procedure at the time of writing.
> We have adjusted the “Advantages of VAEs” section to focus on that VAEs has the two advantages *simultaneously*: (1) expressive representation power as a non-linear model; (2) the number of parameters being independent of the number of users. Not all MF methods has both advantages simultaneously.
>
> Q: On claims of SoTA:
> A: While the survey “A worrying analysis of Recent Neural Recommendation Approaches” does reference these baselines, it’s worth noting that MultiVAE and MultiDAE are reported as exceptions to this trend of poor performance and reproducibility in deep recommender systems. However, we agree that there are strong methods which we do not compare to, and have changed many of the comparative statements to reflect that “RaCT improves strong baselines, rather than achieving SoTA”.
>
> Thanks for pointing out the papers on SVD and others. This is very informative.  We are glad to cite all of them.  In the revision,  we have added (1) a paragraph at the end of the main paper to discuss this; (2) the connection on the importance of user behavior statistics in our critic model to the scaling trick in SVD [Nikolakopoulos 2019].
>
> Q: On the limitation of LVMs:
> A: We clarify that we view increasing the expressivity of LVMs as an important direction of research, although “Neural Collaborative Filtering” is not necessarily an excellent example of this. Therefore, we have reworded our first introductory paragraph to focus on the motivation for applying more expressive models as opposed to making specific performance claims.
>
>
> “Other comments”
> 1) We agree that with a sparse similarity matrix inference can be drastically sped up. However, we are mainly addressing the superior performance of EASE on certain large datasets, which learns a dense representation of the similarity matrix. We have revised this statement for clarity: ``EASE requires to compute a dense item-to-item similarity matrix, making its inference on MSD roughly 30 times more expensive than for VAE or RaCT.’’
> 2) Fixed, thank you.
> 3) Thanks for pointing out the connections to two-level methods and MAB. In the revision, we add a brief discussion in the related work section: ``The proposed actor-critic method belongs to the general two-level architectures for recommendation systems, where a coarse to fine prediction procedure is used.’’
> 4) Fixed, thank you.
> 5) We re-ran the experiments for both VAE and RaCT on the ml-20m dataset with 5 random seeds and found they are both very stable -- the standard deviation of test-set NDCG is roughly 0.0005 for both methods.

---

### Official Review · AnonReviewer1 · 2019-10-23
**Official Blind Review #1**

**Rating:** 6

**Review:**

This paper builds on the line of work of developing latent variable models (LVMs) for collaborative filtering with implicit feedback (e.g. which items a user has previously clicked on).  While variational autoencoders allow convenient construction of nonlinear LVMs, they are trained by maximizing the multinomial likelihood for the items selected.  Thus the training objective is not perfectly matched to the evaluation objective, which is typically something like NDCG @N or RECALL @N, neither of which is differentiable.  This paper proposes an actor-critic RL approach to train the nonlinear LVM directly for the NDCG loss.  The idea is to create a critic model that learns to approximate NDCG, and to train the actor to optimize this approximate objective.  However, they find that they are unable to build an effective critic when taking the predictions and ground truth directly.  However, they find that with a simple set of 3-features, they can build an effective critic that gives improved or competitive performance on several metrics on 3 large-scale datasets.  When their method (RaCT) is not the best, it is beaten by EASE, which is computationally much more expensive.

This work seems to fit naturally into the line of work on nonlinear LVMs.  While porting MLE training to RL training is a standard operation these days, in this case it wasn't entirely trivial -- one couldn't just use REINFORCE and be done.  Introducing features and an actor/critic setup isn't an amazing innovation, but the method works well, has computational benefits, and has compared to many other models on 3 datasets.  The paper is very polished and nice to read so that somebody can really learn from it. I vote for accepting this paper.

Some specific questions and comments:
- In the 'Pitfals of VAEs' section, it says "A preserves the ranking order of the target, while B does not."  I'm seeing that A and B give the same rankings, so I don't understand the comment.
- I don't quite understand the explanation for why REINFORCE wouldn't be effective.  I'm more convinced by the fact that the critic failed to learn when given the full prediction set and ground truth.

**Experience Assessment:**

I have published one or two papers in this area.

**Review Assessment: Checking Correctness Of Derivations And Theory:**

N/A

**Review Assessment: Checking Correctness Of Experiments:**

I assessed the sensibility of the experiments.

**Review Assessment: Thoroughness In Paper Reading:**

I read the paper at least twice and used my best judgement in assessing the paper.

---

> ### Author Response · Authors · 2019-11-14
> **Response to Reviewer #1**
>
> Thank you for your review. First, we apologize for the confusing example of NLL vs NDCG — we have updated the PDF to improve clarity. We replaced the figure with an example that demonstrates NLL assigning a better score to imperfectly-ranked items than to perfectly-ranked ones. In contrast, the ranking-based metrics such as NDCG aligns well with the scoring of items.
>
> Gradient estimation in REINFORCE usually requires a sufficiently large number of samples from the action space, which in our case is a continuous 10K-dimensional distribution. This requires a prohibitively high number of samples to make a reliable gradient update.  Please see the “Discussion and Related Work” section of [*] for good exposition of this point. In contrast, the proposed RaCT is inspired by actor-critic, a well-trained critic can circumvent the high variance problem, by learning an explicit relationship between NDCG and item-score, We have updated the citation in our paper to reflect this more informative source.
>
> [*] Silver, David, et al. "Deterministic policy gradient algorithms." 2014.

---

### Decision · Program_Chairs · 2019-12-19

**Decision:**

Accept (Poster)

**Comment:**

The reviewers generally agreed that the application and method are interesting and relevant, and the paper should be accepted.

I would encourage the authors to carefully go through the reviewers' suggestions and address them in the final.